# Cost-benefit analysis of a multicomponent breastfeeding promotion and support intervention in a developing country

**Maya Basbous**[1], **Nadine Yehya**[2], **Nisreen Salti**[3], **Hani Tamim**[4,5], **Mona Nabulsi**[6]*

1 The Scholars in HeAlth Research Program, Faculty of Medicine, American University of Beirut, Beirut, Lebanon, 2 Department of Public Affairs and Marketing, UC Davis, Davis, CA, United States of America, 3 Department of Economics, Faculty of Arts and Sciences, American University of Beirut, Beirut, Lebanon, 4 Department of Internal Medicine, Faculty of Medicine, American University of Beirut, Beirut, Lebanon, 5 College of Medicine, Alfaisal University, Riyadh, Saudi Arabia, 6 Department of Pediatrics and Adolescent Medicine, American University of Beirut, Beirut, Lebanon

* mn04@aub.edu.lb

## Abstract

### Background

Studies on breastfeeding promotion and support interventions suggest some economic benefits. This study assessed the direct and indirect costs of a multicomponent breastfeeding promotion and support intervention during the first two years of the infant's life.

### Methods

This is a cost-benefit analysis of data generated from a randomized controlled trial that investigated whether provision of a multicomponent breastfeeding promotion and support intervention to Lebanese mothers in the first six months postpartum would improve breastfeeding rates compared to standard obstetric and pediatric care. Data of 339 participants on sociodemographics, mother and infant health, infant nutrition, direct and indirect costs of the intervention were used to assess the benefit-cost ratio (BCR) of the intervention at one, six, 12, and 24 months as primary outcome. Secondary outcomes included overall costs of infant nutrition and infant-mother dyad health costs during the first two years. Multiple linear regression models explored the effect of the intervention on the overall infant nutrition cost and mother-infant health costs. Similar regression models investigated the association between cost variables and infant nutrition types (exclusive breastfeeding, mixed feeding, artificial milk). Intention to treat analyses were conducted using SPSS (version 24). Statistical significance was set at a $p$-value below 0.05.

### Results

The prevalence of *Exclusive/Predominant* breastfeeding among participants declined from 51.6% in the first month to 6.6% at the end of second year. The multicomponent breastfeeding intervention incurred 485 USD more in costs than the control group during the first six months but was cost-efficient at one year (incremental net benefits of 374 USD; BCR =

**Data Availability Statement:** The anonymized data is submitted to PLOS ONE as Supplementary file (S5).

**Funding:** This study was supported by a grant from the Medical Practice Plan, Faculty of Medicine, the American University of Beirut granted to MN. The funding body had no role in the design of the study, and collection, analysis, and interpretation of data, or in writing the manuscript.

**Competing interests:** The authors have declared that no competing interests exist.

2.44), and two years (incremental net benefits of 472 USD; BCR = 2.82). In adjusted analyses, the intervention was significantly associated with fewer infant illness visits in the first year ($p = 0.045$). Stratified analyses by the infant nutrition type revealed that infants who were on *Exclusive/Predominant*, or *Any Breastfeeding* had significantly more favorable health outcomes at different time points during the first two years ($p<0.05$) compared to infants receiving *Artificial Milk* only, with health benefits being highest in the *Exclusive/Predominant* breastfeeding group. Moreover, *Exclusive/Predominant* and *Any Breastfeeding* had significantly lower costs of infant illness visits, hospitalizations, and infant medications during the two years ($p<0.05$) but had additional cost for maternal non-routine doctor visits due to breastfeeding (all $p$ values <0.05). Whereas the overall cost (direct and indirect) during the first six months was significantly lower for the *Exclusive/Predominant* breastfeeding infants ($p = 0.001$), they were similar in infants on *Mixed Feeding* or *Artificial Milk*.

## Conclusions

Breastfeeding is associated with significant economic and infant health benefits in the first two years. In the context of the current economic crisis in Lebanon, this study provides further evidence to policymakers on the need to invest in national breastfeeding promotion and support interventions.

## Introduction

Breast milk is the ideal nutrition for preterm and term infants that ensures healthy growth and development of infants and young children [1]. Breastfeeding (BF) has many nutritional and developmental benefits for children with health benefits extending to their mothers and to the society at large [2–5]. The World Health Organization (WHO) recommends exclusive breastfeeding (EBF) from the first hour after birth until the first six months of life, and the continuation of BF until two years of age with nutritious complementary foods. WHO defines EBF as feeding the infant human milk only without any additional food or drink, not even water [6]. Despite this recommendation, the rates of initiation of BF within the first hour of birth and the rates of EBF in the first two days of life were suboptimal in many countries [7]. Moreover, the overall EBF rate in infants below five months of age in the world averages 48% with the rates in low-middle income countries (LMICs) varying between 32% and 61% [7].

In Lebanon, the prevalence of EBF is low, estimated at 58.3% in infants less than one month and 15% in infants below five months of age [8]. Moreover, the rates of early BF initiation and BF continuation until one year of age were low at 41% and 27.1%, respectively [9, 10]. Studies from Lebanon identified several barriers to the practice of BF, such as societal misconceptions about BF, the lack of lactation specialists, the lack of social support for breastfeeding mothers, and the absence of national programs that promote BF practices on maternity wards or in the community [11–13]. The Lebanese government led efforts to improve BF practices in the country through establishing a national committee on the nutrition of infants and young children, organizing national campaigns to promote BF, regulating the marketing of breast milk substitute products for infants through law No. 47/2008, and introducing the Baby-Friendly Hospital Initiative (BFHI). Despite these efforts, the rates of BF did not rise, and only 7% of Lebanese hospitals were designated as baby-friendly [14].

In response to the low national breastfeeding rates and societal barriers to breastfeeding, we conducted a multicenter, two parallel-arm, single-blind multicomponent BF promotion and support trial that compared the effect of provision of a package consisting of prenatal BF education, professional lactation support and peer support for up to six months postpartum on six-month EBF rate of women in the community, in comparison to standard medical care. The multicomponent BF intervention resulted in a significantly higher EBF rate in the experimental group compared to the control group (adjusted OR = 2.02; 95% CI: 1.20–3.39) [15]. This study aims to assess the overall costs (direct and indirect) and benefits of the multicomponent BF intervention at different time points during the first two years of the infant's life. It is a timely study as Lebanon is currently suffering from a severe economic crisis, with increasing rates of poverty and rapid drops in the purchasing power for large segments of its population [16]. While the effectiveness of the intervention on promoting EBF was already established in our trial [15], providing evidence that the intervention is also cost effective is essential for any scaling up of such an intervention. As the country loses ground on many other factors protecting child health (immunization, access to healthcare and medication, access to clean water) because of the economic crisis, identifying cost-effective interventions that impact protective behavioral factors such as breastfeeding is even more crucial. Evidence from previous economic evaluations of BF promotion and/or support interventions that were conducted in developing and developed countries reveal that BF may have economic benefits. However, none of the studies looked at whether the economic effect persisted beyond the initial few months postpartum, took into consideration all childhood and maternal illnesses, or included a detailed breakdown of the direct and indirect costs related to BF [17–20]. We hypothesized that breastfeeding's economic and infant-mother health benefits persist through the first two years. We therefore conducted this cost-benefit study to bridge the existing knowledge gap by assessing the direct and indirect costs of our multicomponent BF intervention at one, three, six, twelve, and 24 months postpartum, by including infant and maternal illnesses during these periods.

## Materials and methods

### Study design

This is a cost-benefit analysis of data generated from a multi-center, two-arm, single-blind randomized controlled trial (RCT) that aimed to investigate whether the provision of a multicomponent BF promotion and support intervention to mothers in the community during the first six months postpartum would improve BF rates, as compared to standard obstetric and pediatric care. The trial was conducted between December 6, 2013, and January 21, 2016. The trial was approved by the Institutional Review Board of the American University of Beirut, reference number PED.MN.08. Written informed consent was obtained from all participants. The intervention consisted of three components: prenatal BF education, peer support, and professional lactation support. Details of the trial procedures and findings are described in the trial protocol [21] and report [15]. Briefly, prenatal BF education consisted of at least one antenatal BF education session, provision of an educational pamphlet, and video about BF to raise participants' BF knowledge. Peer support was provided to the participants by trained lay mothers from the community via scheduled telephone calls and hospital/home visits for six months postpartum to strengthen the mothers' social networks and support. Professional lactation support was provided by certified lactation experts through postpartum hospital and home visits for six months to improve maternal BF skills and efficacy. The control group received standard postpartum obstetric and pediatric care provided by hospital nurses and pediatricians.

## Participants

Participants were mothers who participated in the breastfeeding trial and completed at least one month of follow up (n = 339). The trial had 90% power to detect a 10% difference in the 6-month EBF rate between the intervention and control groups, with 5% type 1 error and a potential 30% attrition rate. The sample size was estimated at 443 participants that were recruited from the antenatal clinics of two healthcare centers in Beirut, Lebanon. After obtaining written informed consent, participants were randomly assigned to an experimental group (n = 160) and a control group (n = 179). Inclusion criteria were healthy pregnant women who were in their first or second trimester of pregnancy and who planned to breastfeed after delivery. Women with any chronic medical condition, abnormal fetal screen, twin gestation, no intention to breastfeed, delivery prior to 37 weeks of gestation, were past the second trimester of pregnancy, or residing outside Lebanon were excluded [15, 21].

## Definitions

In this cost-benefit analysis study, we grouped the infants according to the type of milk they received for nutrition during the first six months of life. Nutrition group types were *Exclusive Breastfeeding*, *Predominant Breastfeeding*, *Mixed Feeding*, *Any Breastfeeding*, *Exclusive Formula Feeding* (EFF), and *Any Formula Feeding* (Any FF). These groups are not mutually exclusive.

- *EBF* was defined as feeding the infant human milk only without any additional food or drink, not even water [6].

- *Predominant Breastfeeding* is feeding the infant human milk as the predominant source of nutrition, but allowing water, water-based drinks, or a maximum of two feedings of artificial milk in one week during the first six months.

- *Mixed Feeding* is feeding the infant both human milk and more than two feedings of artificial milk per week.

- *Any BF* is feeding the infant human milk with or without artificial milk. This group includes EBF, Predominant Breastfeeding and Mixed Feeding.

- *Any FF* is feeding the infant artificial milk with or without human milk. This group includes EFF, Predominant Breastfeeding and Mixed Feeding.

- *EFF* is feeding the infant artificial milk exclusively.

## Data collection

Data on the following variables were collected at months 1, 3, 6, 12, and 24 postpartum: 1) maternal variables: participant's allocation (intervention vs. control), age, employment (Yes/No), number of living children, education (less than university vs. university degree), household monthly income, maternal non-routine doctor visit relating to BF (Yes/No), number of maternal non-routine doctor visits due to BF, cost of maternal non-routine doctor visits due to BF; 2) infant variables: type of infant nutrition (*Exclusive/Predominant Breastfeeding*, *Mixed Feeding*, *EFF*, *Any FF*, or *Any BF*), number of artificial milk bottles per day, volume of artificial milk per bottle, brand of artificial milk formula, price of one can of formula, number of formula cans consumed per week, brand of water bottles used to prepare bottles of artificial milk, price of one water bottle, number of water bottles consumed per week, infant's illness visit to a

physician (Yes/No), number of infant illness visits, diagnosis of infant illness at each visit, cost of each illness visit, third party payment for illness visit (Yes/No), use of drugs for infant illness (Yes/No), cost of drugs per illness, infant hospitalization for an illness (Yes/No), number of infant hospitalizations, diagnosis of infant illness at each hospital admission, cost of each hospital admission, third party payment for infant hospital admission (Yes/No), and cost of drugs used during/after hospitalization. Trained research assistants collected baseline and follow-up data, including nutrition, health, and cost data using standardized data collection forms. Nutrition data were adjudicated by lactation consultants. The primary investigator conducted weekly meetings with the research team to review the accuracy of collected data. Mothers were contacted for missing data or whenever data needed further scrutinization.

## Sample size

The trial recruited 446 mothers with 222 participants allocated to the intervention group and 224 to the control group. Of these, 339 participants had available data at one month and hence constituted the sample of this cost-benefit analysis study. Fig 1 summarizes the participants flow during the first 24 months of the trial.

## Outcomes

The primary outcome of the study was the benefit-cost ratio (BCR) of the multicomponent BF intervention at one month, six months, 12 months, and 24 months. The BCR compares the discounted benefits of the intervention against its discounted implementation costs at each time point.

Secondary outcomes included the differences between the intervention and control groups in terms of: 1) the overall cost of infant nutrition, 2) the overall health costs for infant and mother (summation of the costs of infant illness visits, infant hospitalizations, drugs used to treat infants, and maternal non-routine consultations due to BF), during the following time periods: first month; months two to three; months four to six; first six months together; months 7 to12; first 12 months together (first year of infant's life); months 13 to 24 (second year of infant's life); and all 24 months together (first two years of infant's life). Other secondary outcomes were the differences in the overall cost of infant nutrition and overall health costs of the infant-mother dyad at the same time periods stratified by the type of infant nutrition, irrespective of maternal allocation in the trial.

## Statistical analysis

We summarized continuous variables as means and standard deviations (SD) and categorical variables as counts and proportions. To calculate the BCR, we first calculated the Net Present Value (NPV) of expenditures incurred by each participant in the trial (control and intervention groups). To calculate the NPV, all expenses were discounted using an annual discount rate (r) of 3%, which is appropriate for developing countries based on recommendations for economic analyses [22, 23]. The NPV represents the present-day equivalent of the value of the stream of expenditures over the duration of the observation period, considering the time value of money (e.g., spending 100 USD today is not equivalent to spending 100 USD one year from today) [22, 24]. Assuming that the direct and indirect expenditures were spent uniformly by month, we used a monthly discount rate (r/12). Therefore, the NPV of the overall total cost for

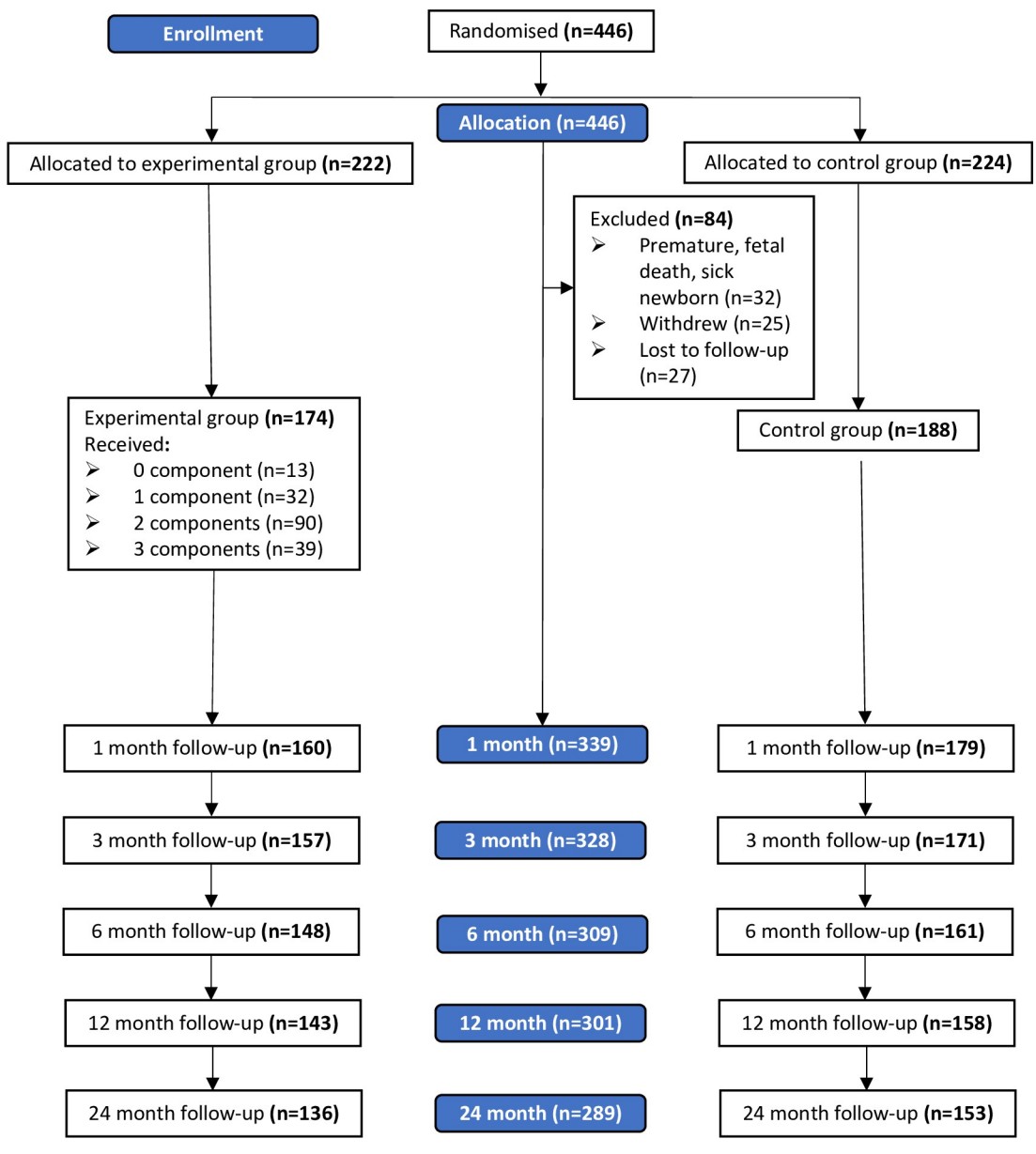

**Fig 1. Participants' flow in the trial.**

a specific month is equal to:

$$\frac{Mean\ total\ cost\ during\ a\ specific\ month}{Discount\ Factor}$$

The Discount Factor for any expense that happens *i* months from today is equal to: $\left(1 + \frac{r}{12}\right)^{i}$.

We calculated the cost difference between the experimental and control groups as follows: cost difference = mean net present value per participant in the control group–mean net present value per participant in the experimental group. The net benefits of the intervention were calculated as the cost difference–net implementation costs of the intervention. Therefore, the

formula of the BCR is as follows:

$$\frac{Mean\ NPV\ of\ total\ cost\ per\ participant\ in\ the\ control\ group\ -\ Mean\ NPV\ of\ total\ cost\ per\ participant\ in\ the\ experimental\ group}{NVP\ of\ the\ net\ implementation\ cost\ per\ participant\ in\ the\ experimental\ group}$$

A BCR greater than 1 indicates that the intervention of interest is efficient and saves money relative to the control. A ratio less than 1 indicates that the intervention is inefficient, and costs more than it saves relative to the control.

We conducted multiple linear regression analyses to compare the intervention and control groups with respect to the overall cost of infant nutrition, and the overall health costs for infant and mother, adjusting for any baseline characteristic that was significantly different between the two trial groups. These variables were the the number of children and the monthly household income, both of which were entered in the regression models as predicros to adjust for their confounding effects. These comparisons were done separately for month one, months two to three, months four to six, all first six months, months 7 to12, all first 12 months, months 13 to 24, and all 24 months. Moreover, we compared differences in infant illness visits and hospitalizations between the intervention and control groups as dichotomous (Yes/No) and continuous variables (frequency) using logistic and linear regressions, respectively. Similarly, we compared these two variables in *EBF* vs. *Mixed Feeding* vs. *EFF* groups using Chi-square and ANOVA for dichotomous and continuous variables, respectively.

With respect to the comparisons of infant nutrition costs and health costs of the infant-mother dyads according to the infant nutrition group, we compared *EBF* vs. *Mixed Feeding* vs. *EFF* using ANOVA; *EBF* vs. *Any FF*, and *Any BF* vs. *EFF* using the independent Student's *t*-test. All statistical analyses were conducted as intention to treat analyses using SPSS (version 24). Statistical significance was evaluated at a *p*-value below 0.05.

## Results

### Baseline characteristics

The flow of the participants in the breastfeeding trial is shown in Fig 1. Of the 339 participants who completed one month of follow up in the trial, 50 participants either withdrew or were lost to follow up by the end of the second year of trial. The baseline characteristics of these 50 participants were not significantly different from the characteristics of those who continued in the trial. The participants had a mean (*SD*) age of 29.5 (4.9) years, with more than half (*n* = 176, 52%) being employed, having a university-level education (*n* = 267, 79%), with at least one child (*n* = 181, 53.4%), and a monthly household income above 1,000 USD (*n* = 229, 72%). A significantly higher proportion of women in the trial's intervention group were primigravid (*p*<0.001) and had a higher monthly household income (*p* = 0.003). Otherwise, the two groups had similar baseline characteristics (Table 1).

### Cost-benefit analysis

The implementation cost of the multicomponent intervention was allocated to the intervention group only. It included compensation for the lactation consultants (31,200 USD), telephone charges and commuting of the support mothers (5,820 USD), salaries of three research assistants (93,600 USD), the cost of the BF video production (8,500 USD), BF booklets and pamphlets (1,500 USD), and the research assistants' commuting cost (1,500 USD). The implementation cost per participant was 43.60 USD in the first month, and 261.61 USD for each of the following time periods: 0 to 6 months (first six months of infant's life), 0 to 12 months (first year of infant's life), and 0 to 24 months (first two years of infant's life).

**Table 1. Baseline demographic characteristics in the trial's experimental and control groups.**

|  | Total (N = 339) | Control (n = 179) | Experimental (n = 160) | *p*-Value |
|---|---|---|---|---|
| **Continuous Variables** | *Mean (SD)* | *Mean (SD)* | *Mean (SD)* |  |
| **Age (years)** | 29.5 (4.9) | 29.3 (5.2) | 29.6 (4.6) | 0.550 |
| **Categorical Variables** | *n (%)* | *n (%)* | *n (%)* |  |
| **Employment status** |  |  |  |  |
| **Homemakers** | 163 (48.1) | 90 (50.3) | 73 (45.6) | 0.392 |
| **Employed** | 176 (51.9) | 89 (49.7) | 87 (54.4) |  |
| **Number of living children** |  |  |  |  |
| **0** | 158 (46.6) | 65 (36.3) | 93 (58.1) | **<0.001** |
| **1+** | 181 (53.4) | 114 (63.7) | 67 (41.9) |  |
| Education |  |  |  |  |
| ≤ High school | 72 (21.2) | 44 (24.6) | 28 (17.5) | 0.112 |
| University | 267 (78.8) | 135 (75.4) | 132 (82.5) |  |
| Monthly household income (USD)(missing data = 22) |  |  |  |  |
| ≤ 1000 | 88 (27.8) | 59 (34.7) | 29 (19.7) | **0.003** |
| > 1000 | 229 (72.2) | 111 (65.3) | 118 (80.3) |  |

In the unadjusted intention to treat analyses, the intervention group had a significantly lower proportion of infants needing hospital admission during the period 7 to 12 months (5.6% in the intervention vs. 12.7% in the control group; $p = 0.035$), and a smaller number of doctor visits for infant illnesses during the first 12 months (intervention: 2.22 ± 2.30, control: 22.97 ± 3.10; $p = 0.016$). However, in the adjusted intention to treat analyses, only the number of doctor visits for infant illnesses in the first 12 months remained statistically significant ($p = 0.045$). No other significant differences were found between the two groups at other time points with respect to infant and mother health outcomes, or to the cost of the following: infant nutrition, doctor visits for infant illnesses, infant hospitalizations, infant medications, maternal non-routine doctor visits due to BF, and overall total costs in both unadjusted and adjusted analyses (S1 File).

The results of the cost-benefit analysis of the intervention are shown in Table 2 at the following time periods: first month, first six months, first year, and first two years of the infant's life. After applying a discount rate of 3%, the intervention incurred 138 USD more in overall costs per participant in the first month, and 485 USD more in overall costs per participant during the first six months, relative to the control group. However, the intervention was cost-efficient by the end of the first year (incremental net benefits of 374 USD; BCR = 2.44), and by the end of the first two years (incremental net benefits of 472 USD; BCR = 2.82).

## Comparisons by the type of infant nutrition

The prevalence of *Exclusive/Predominant BF* among trial participants decreased from 51.6% by the end of first month to 6.6% by the end of the second year. Similarly, *Mixed Feeding*

**Table 2. Findings of the cost-benefit analysis in the experimental and the control groups.**

|  | Month 0–1 | Month 0–6 | Month 0–12 | Month 0–24 |
|---|---|---|---|---|
| **Cost difference** | -94.88 | -225.40 | 633.33 | 730.99 |
| **Net Implementation Costs** | 43.49 | 259.34 | 259.34 | 259.34 |
| **Net Benefits** | -138.37 | -484,74 | 373.99 | 471.65 |
| **Benefit Cost Ratio** | - | - | 2.44 | 2.82 |

prevalence decreased from 43.1% to 1.4% during the same period. In contrast, prevalence of *EFF* increased from 5.3% at month one to reach 92% by the end of two years (Table 3).

**Exclusive/Predominant BF vs. Any FF.** Infants who were on *Exclusive/Predominant BF* had more favorable health outcomes at different time points during the first two years than infants who were on *Mixed Feeding* or *EFF*. The proportion of infants with illness-visits and hospitalizations in the first month was significantly smaller in the *Exclusive/Predominant BF* group (14.9% vs. 26.9%, $p = 0.007$; 3.4% vs. 8.5%, $p = 0.035$; respectively). Moreover, this group had fewer illness visits in the first three months, and fewer hospitalizations during the first year (all $p$ values $<0.05$). The overall cost (direct and indirect) was significantly lower in the *Exclusive/Predominant BF* group during the first six months ($p = 0.001$). As expected, the cost of water and formula in the *Any FF* group was significantly higher at all time points, whereas the costs related to infant health were significantly lower in the *Exclusive/Predominant BF* group such as illnesses visits in the first month ($p = 0.001$), infant hospitalization during the first six months ($p = 0.016$) and the second year ($p<0.001$), infant medications during the first month ($p = 0.017$), months 2 and 3 ($p = 0.032$), and second year ($p<0.001$). Cost of maternal non-routine doctor visits due to BF was significantly higher in the *Exclusive/Predominant BF* group only during months 4 to 6 ($p = 0.031$). Details of the comparisons between the two groups are summarized in S2 File.

**Any BF vs. EFF.** When comparing the groups of *Any BF* and *EFF*, *Any BF* was associated with a significantly lower proportion of infants who had illness visits (24.4% vs. 36.5%, $p = 0.04$) and hospitalizations (2.4% vs. 10.8%, $p = 0.004$) during the second and third months, and fewer hospitalizations during the first six months (10.6% vs. 29.4%, $p = 0.035$). Similarly, *Any BF* was significantly associated with fewer number of infant hospitalizations during second and third months ($p = 0.027$), and during second year of the infant's life ($p<0.001$). As for costs, *EFF* had significantly higher costs for water and formula at all time points. However, *Any BF* was associated with significantly lower costs of infant hospitalizations during the second and third months ($p = 0.038$) and during the second year ($p<0.001$), as well as lower cost of infant medications during the second year ($p<0.001$). *Any BF* was associated with higher cost of maternal non-routine doctor visits due to BF during months 4 to 6 ($p = 0.047$), in the first six months ($p<0.001$), and in the first year ($p<0.001$). However, the overall cost was not significantly different between the two groups at any time point. Further details of the comparisons between *Any BF* and *EFF* groups are shown in S3 File.

**Exclusive/Predominant BF vs. Mixed feeding vs. EFF.** The comparison of the three groups *Exclusive/Predominant BF*, *Mixed Feeding*, and *EFF* showed a significant difference in the proportion of infants who had illness visits in the first month being lowest in the *Exclusive/Predominant BF* group and highest in the *EFF* group (14.9% vs. 25.3% vs. 38.9%, $p = 0.009$). A similar trend was observed in the proportion of hospitalized infants (3.4% vs. 7.5% vs. 6.7%, $p = 0.039$) in the first month, and during the first six months (6.9% vs. 12.2% vs. 29.4%,

**Table 3. Types of infant nutrition at different time points during the first two years.**

|  | Type of Infant Nutrition | | | |
|---|---|---|---|---|
|  | EBF/Predominant BF | Mixed Feeding | EFF | Total |
| TIME POINTS | n (%) | n (%) | n (%) | N |
| Month 1 | 175 (51.6) | 146 (43.1) | 18 (5.3) | 339 |
| Months 2 and 3 | 140 (42.7) | 114 (34.8) | 74 (22.6) | 328 |
| Months 4 to 6 | 105 (34) | 81 (26.2) | 123 (39.8) | 309 |
| Months 7 to 12 | 68 (22.6) | 49 (16.3) | 184 (61.1) | 301 |
| Months 13 to 24 | 19 (6.6) | 4 (1.4) | 264 (92) | 287 |

$p = 0.033$), and in the number of infant illness visits in the first month ($p = 0.004$), number of infant hospitalizations at one month ($p = 0.041$) and during the first six months ($p<0.001$). Healthcare costs were lowest in the *Exclusive/Predominant* group and highest in the *EFF*, specifically in infant illness visits at one month ($p<0.001$), infant hospitalization during the first six months ($p = 0.001$), and overall cost during the first six months ($p<0.001$). The cost of water and formula was highest in the *EFF* group and lowest in the *Exclusive/Predominant BF* group at all time points. Further details of the comparisons among the three groups are shown in S4 File.

## Discussion

In this study, we conducted a cost-benefit analysis of a multicomponent BF promotion and support intervention composed of peer support, professional lactation support, and BF education. We found that although the intervention was not cost-efficient in the first six months postpartum, it was cost-efficient in years one and two. The higher intervention cost in the first six months can be due to the high number of home visits, costs of professional lactation support and BF education delivered to mothers in the intervention group. The net monetary benefits of the intervention were 2.44 times, and 2.82 times the costs of its implementation in year one and year two, respectively.

There are few studies in the literature that evaluated the cost-effectiveness of breastfeeding promotion interventions reporting conflicting results [5, 17–19, 25]. The differences in the cost-effectiveness of the breastfeeding interventions reported in these studies may be due to differences in their BF promotion interventions, study designs, durations of intervention delivery, and in the outcomes used to report cost-effectiveness. Chola et al. [17] assessed the cost-effectiveness of community-based peer counseling to promote BF and reduce diarrhea-associated mortality and morbidity in Uganda, as compared to health facility-based BF promotion. Home-based peer counseling doubled the 3-month BF prevalence but was not cost-effective in averting diarrhea-associated disability adjusted life years (DALYs) and had an incremental cost of 137 USD per mother-child pair at six months in comparison to the health facility-based support. In contrast, our multicomponent BF promotion intervention had a higher incremental cost of 485 USD per participant during the first six months, which is explained by the additional costs of professional lactation support and BF education delivered to mothers in the intervention group. However, our multicomponent intervention was cost-beneficial in years one and two. To note, there are several differences between our study and that of Chola et al. Whereas we prospectively collected data on all infant morbidities and maternal BF-associated morbidities over two years, Chloe et al. estimated infant diarrhea-related costs based on reports from previous studies, and for the first six months only. Moreover, our intervention had three BF promotion and support components and not only peer support.

Frick et al. [19] conducted a cost-benefit analysis of a breastfeeding support intervention delivered by nurses and peer counselors (Breastfeeding Support Team) to low-income breastfeeding women in Baltimore, Maryland. The intervention included hospital visits and three home visits by a community health nurse/peer counselor, in addition to telephone contacts. The economic evaluation focused on whether the costs of intervention personnel (direct costs only) would be offset by reducing health costs and formula costs. They found that the direct intervention costs were partially offset by reducing medical care utilization and formula feeding costs at weeks 4 to 8. The analysis, however, did not include indirect costs and was limited to the first 12 weeks of the intervention. Walters et al. [5] analyzed the cost-benefit of a modeled breastfeeding promotion strategy implemented in Vietnam at a national level and the costs of NOT breastfeeding in seven countries in Southeast Asia. The modeled national

breastfeeding promotion strategy had a BCR of $2.39 for every $1 invested, which was equivalent to a 139% return on investment. Moreover, this national strategy was estimated to prevent 200 child deaths per year in Vietnam. Bland et al. [25] conducted a non-randomized trial in a high HIV prevalence area in South Africa to promote EBF for six months by offering home-based BF counselling. They reported high EBF rates among mothers who received the intervention, as well as low rates of mastitis and other breastfeeding-associated health issues, and lower risk of postnatal HIV transmission. The economic evaluation of the home-based BF counselling was assessed by Desmond et al. [18] under three different scenarios, if it were implemented at provincial level: 1) implementation of the full intervention as in the protocol (rural areas: 10.4 home visits, urban areas: 10.4 clinic visits); 2) implementation of a simplified intervention with less frequent pre-and post-natal visits and more clinic-based visits (rural: 4.2 home visits and 2 clinic visits, urban: 2 home visits and 4.2 clinic visits); 3) implementation of a basic intervention which is entirely clinic-based (3 clinic visits for both rural and urban cohorts). The simplified scenario was found to be the most efficient in terms of cost per additional month of EBF. The cost-efficiency of the simplified scenario may be due to the lower number of visits as compared to our intervention, which consisted of more home visits (days 1, 3, 7, and 15 post-hospital discharge, and then monthly for six months postpartum). Hence, the higher number of home visits in our intervention explains the lack of cost-efficiency in the first six months.

Our intervention was also associated with a reduced number of infant hospitalizations and doctor illness visits during the first year of life. However, this was not paralleled with reductions in the costs of infant nutrition, infant illness visits, infant hospitalizations and medications, or maternal non-routine doctor visits due to BF at any time point. Moreover, the difference in the overall total costs between the intervention and the control groups was not statistically significant. This may be attributed to the fact that only 22.4% of the participants in the intervention arm complied with the three components of the intervention, which may have attenuated the association between the breastfeeding intervention and health-related costs [15]. In addition, the two trial groups differed in two baseline characteristics, namely parity, and socioeconomic status that may potentially affect their compliance with the experimental intervention. Wealth disparities have been shown to affect BF continuation, with poorer women being 1.5 times more likely to BF for two years as compared to richer ones [26]. Similarly, the six-month BF rate is less in first-time mothers than mothers who have at least one child (59% vs. 69%, p<0.001) [27]. In our cohort, more women in the intervention group were primiparous and had a higher monthly income. These differences may have contributed to less BF in this group and to the attenuation of the differences in the overall total costs between the two groups.

Our study also revealed that, irrespective of maternal allocation in the trial, infants who were on *Exclusive/Predominant BF* had less infant illness visits and hospitalizations, as well as less health-related costs and overall cost in the first six months of their lives. For example, in the first three months, we found a significant reduction in infant illness visits, hospitalizations, and medication cost. Similarly, there was a reduction in infant hospitalizations and medication cost between 12 and 24 months. However, *Exclusive/Predominant BF* was associated with an increase in the cost of maternal non-routine visits due to BF between the 3rd and the 6th months. This is an expected finding as BF women may experience BF-associated complications, such as mastitis, engorgement, or sore nipples necessitating medical care. Interestingly, infants on any BF also showed significant reductions in infant illness visits, hospitalizations, and associated costs at different time points in the first two years, and a similar increase in the cost of maternal non-routine doctor visits. These observed BF benefits in infants' health outcomes and its associated cost savings are consistent with the literature. In a meta-analysis by Victora et al.

[2], breastfeeding was shown to be associated with significant protection against child infections and other disorders, as well as reduced maternal risk for breast and ovarian cancers, and type 2 diabetes. Optimal breastfeeding may decrease under-five child mortality by 823,000 deaths and prevent 20,000 maternal deaths from breast cancer each year. A cost-analysis study on the burden of suboptimal breastfeeding in the United States concluded that $13 billion could be saved and 911 excess infant deaths could be prevented if 90% of US mothers exclusively breastfeed for six months [28]. Another cost analysis study from the UK estimated that supporting mothers to EBF for four months would save at least £11 million annually, and that doubling the proportion of mothers breastfeeding for at least seven months would reduce maternal breast cancer incidence and save at least £31 million each year, at 2009–2010 value [3].

## Strengths and limitations

Our findings are conservative in that we are missing many of the benefits of BF. The fact that our study stopped at two years means we only captured some of the benefits of BF. To the extent that the immunity acquired through BF continues to protect the child past age two years, we are unable to measure that benefit. Another limitation is that the benefits are measured in terms of physical health outcomes (absence of illness), whereas some of the benefits may be cognitive and behavioral. A further limitation is the fact that trial participants were recruited from two private hospitals in Beirut and were women who intended to BF. Therefore, our results may not be generalizable to women in rural areas or to women who did not intend to breastfeed. Moreover, only a small percentage of women complied with the three components of the intervention, which may attenuate the difference in the cost-benefit analysis between the two trial groups, especially that it was an intent-to treat analysis. It should also be noted that our sample suffered from attrition, with a total of 50 participants out of 339 withdrawing before month 24. Attrition is not uncommon in studies with long follow up, however, it is somewhat reassuring that the rate of attrition in this study is balanced across the two groups. We cannot exclude the potential for information bias since the costs of infant nutrition and healthcare utilization were self-reported and not based on invoices provided by the mothers. Finally, this study did not collect data on maternal food consumption, maternal health problems not related to BF, use and cost of breast milk pumps, and the cost of maternal absenteeism from work. The main strength of our study is its prospective design which reduces recall bias relating to the accuracy of data on infant nutrition and healthcare cost over the first two years of infant's life. Also, we collected data on all infant non-routine health visits as well as maternal health care visits relating to BF-associated problems.

## Conclusions

Our multicomponent BF promotion and support intervention was associated with infant health benefits as well as economic benefits that were noted during the first two years postpartum. Moreover, exclusive, or partial BF was associated with significant infant-health and economic benefits in the first six months postpartum. Hence, our findings suggest that any intervention that increases the rate or duration of BF may be cost-beneficial. In the context of the current economic crisis in Lebanon, this study provides further evidence to policymakers on the need to invest in BF promotion and support interventions in Lebanon.

## Supporting information

**S1 File. Comparison of the trial experimental and the control groups at different time points.**
(PDF)

**S2 File. Comparison of the Exclusive/Predominant BF and any formula groups at different time points.**
(PDF)

**S3 File. Comparison of the exclusive formula feeding and any breastfeeding groups at different time points.**
(PDF)

**S4 File. Comparison of the Excusive/Predominant BF, mixed feeding, and exclusive formula feeding groups.**
(PDF)

**S5 File. Anonymized dataset.**
(XLSX)

## Acknowledgments

We are thankful to all our participants for their valuable contribution to our study. Author Maya Basbous would like to acknowledge the training received under the Scholars in HeAlth Research Program (SHARP) that was in part supported by the Fogarty International Center and Office of Dietary Supplements of the National Institutes of Health (Award Number D43 TW009118). The content is solely the responsibility of the authors and does not necessarily represent the official views of the National Institutes of Health.

## Author Contributions

**Conceptualization:** Nadine Yehya, Hani Tamim, Mona Nabulsi.

**Data curation:** Mona Nabulsi.

**Formal analysis:** Maya Basbous, Nadine Yehya, Nisreen Salti, Hani Tamim, Mona Nabulsi.

**Funding acquisition:** Mona Nabulsi.

**Investigation:** Maya Basbous, Nadine Yehya, Nisreen Salti, Hani Tamim, Mona Nabulsi.

**Methodology:** Maya Basbous, Nadine Yehya, Nisreen Salti, Hani Tamim, Mona Nabulsi.

**Project administration:** Mona Nabulsi.

**Resources:** Mona Nabulsi.

**Software:** Hani Tamim, Mona Nabulsi.

**Supervision:** Mona Nabulsi.

**Validation:** Mona Nabulsi.

**Visualization:** Mona Nabulsi.

**Writing – original draft:** Maya Basbous, Nadine Yehya, Nisreen Salti, Hani Tamim, Mona Nabulsi.

**Writing – review & editing:** Maya Basbous, Nadine Yehya, Nisreen Salti, Hani Tamim, Mona Nabulsi.

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
