## [Decision Letter · Decision Letter 0]

30 Apr 2024

PONE-D-23-37306Cost-benefit analysis of a multicomponent breastfeeding promotion and support intervention in a developing countryPLOS ONE

Dear Dr. Nabulsi,

Thank you for submitting your manuscript to PLOS ONE. After careful consideration, we feel that it has merit but does not fully meet PLOS ONE’s publication criteria as it currently stands. Therefore, we invite you to submit a revised version of the manuscript that addresses the points raised during the review process.

Please submit your revised manuscript by Jun 14 2024 11:59PM**.** If you will need more time than this to complete your revisions, please reply to this message or contact the journal office at plosone@plos.org. Please include the following items when submitting your revised manuscript:A rebuttal letter that responds to each point raised by the academic editor and reviewer(s). You should upload this letter as a separate file labeled 'Response to Reviewers'.A marked-up copy of your manuscript that highlights changes made to the original version. You should upload this as a separate file labeled 'Revised Manuscript with Track Changes'.An unmarked version of your revised paper without tracked changes. You should upload this as a separate file labeled 'Manuscript'.

We look forward to receiving your revised manuscript.

Kind regards,

Kahsu Gebrekidan

Academic Editor

PLOS ONE

Reviewers' comments:

Reviewer's Responses to Questions

**Comments to the Author**

1. Is the manuscript technically sound, and do the data support the conclusions?

Reviewer #1: Yes

Reviewer #2: Yes

2. Has the statistical analysis been performed appropriately and rigorously? 

Reviewer #1: Yes

Reviewer #2: Yes

3. Have the authors made all data underlying the findings in their manuscript fully available?

Reviewer #1: Yes

Reviewer #2: Yes

4. Is the manuscript presented in an intelligible fashion and written in standard English?

Reviewer #1: Yes

Reviewer #2: Yes

5. Review Comments to the Author

Reviewer #1: The authors intended to assess the costs of a breastfeeding promotion and support intervention in the first two years of the infant’s life based on a randomized controlled trial. They analyzed data from 339 participants and concluded that breastfeeding is significantly associated with economic and infant health benefits.

1. Line 141. Please define abbreviation EBF before use it.

2. Line 148. Any BF is defined as feeding the infant human milk with or without artificial milk. But when without artificial milk, how to distinguish it from EBF or predominant breastfeeding?

3. Line 149. How to distinguish EFF from any FF if “any FF” is without human milk.

4. Line 218. There are six groups but why only conduct the analysis to compare three of them?

Reviewer #2: Thank you for this valuable evidence about the need to invest in BF promotion and support interventions in Lebanon.

Abstract:

The abstract is too long. You can shorten the method section specifically the description of the regression.

Introduction:

1. Please elaborate the significance of the study

2. Please add the hypothesis for your study.

METHOD:

Participants:

1. Please report the sample size calculations method

Definitions

Please transfer it to the introduction section.

Data collection

Please report the validity and the reliability of the data collection method and the outcomes.

Line 154: please clarify the education levels.

Statistical analysis

Did you assess the assumption of the statistical test. Are there any violations s to these assumptions??

Baseline characteristics

Line 231: what you did to control the effect of these differences statistically?

Comparisons by the type of infant nutrition

Please give more details about the type of statistical analysis name and variables included in each test.

Discussion:

Line 343: please cite the few studies in the literature.

Line 340: explain your findings. Why did you have such results??

Line 375: cite the Blanded et al. directly after the author not at the end of the sentence.

Line 379: “ this intervention ” which intervention???

Line 422 to 430: transfer to the introduction. And keep a summary for it here.

Limitations:

You need to consider the effect of the sample attrition.

6. PLOS authors have the option to publish the peer review history of their article (what does this mean?). If published, this will include your full peer review and any attached files.

Reviewer #1: No

Reviewer #2: No

---

## [Author Response · Author response to Decision Letter 0]

10 Jun 2024

Date: June 8, 2024 

Dear Editor,

We would like to submit our revised manuscript [PONE-D-23-37306] entitled “Cost-benefit analysis of a multicomponenet breastfeeding promotion and support intervention in a developing country” for publication in PLOS ONE. All changes have been highlighted in yellow in the revised tracked copy. We are very thankful for the constructive comments of the reviewers which we hope will improve our manuscript. 

Below, please find our response to each comment.

Journal Requirements:

1. Please ensure that your manuscript meets PLOS ONE’s style requirements, including those for file naming.

Answer: Done. The manuscript was revised in accordance with PLOS ONE’s style requirements, including file naming. 

2. Data Availability: Please describe the changes you make to Data Availability Statement in your cover letter.

Answer: We submitted our anonymized dataset to PLOS ONE with the revised manuscript as Supporting Information File (S5 File). Kindly update our Data Availability statement on our behalf to reflect this information.

3. Minimal data set: Please upload your study’s minimal data set as either Supporting Information files or to a stable public repository.

Answer: We uploaded our minimal deidentified data set as Supporting Information file (S5 File). 

4. Please include your full ethics statement in the Methods section of your manuscript file. Please includ the full name of the IRB who approved your study, as well as whether or not you obtained written or verbal consnet. 

Answer: We included the following ethics statement in the Methods section: 

“The trial was approved by the Institutional Review Board of the American University of Beirut, reference number: PED.MN.08. Written informed consent was obtained from all participants”.

Response to Reviewer 1: 

Thank you for your constructive comments and the valuable review that helped us improve our manuscript. Please find below our reply to each comment.

1. Line 141. Please define abbreviation EBF beofre use it.

Answer: In the Introduction section, lines 69-72, we added the following sentences as definition of EBF: 

“The World Health Organization (WHO) recommends exclusive breastfeeding (EBF) from the first hour after birth until the first six months of life, and the continuation of BF until two years of age with nutritious complementary foods. WHO defines EBF as feeding the infant human milk only without any additional food or drink, not even water [6]”. 

2. Line 148. Any BF is defined as feeding the infant human milk with or without artificial milk. But when without artificial milk, how to distinguish it from EBF or predominant breastfeeding?

Answer: The various nutrition groups defined are not mutually exclusive. The “Any BF” includes all groups except “EFF”. This was done to allow for a “dichotomization” of the nutrition groups in the analysis. Comparison of “Any BF” versus “EFF” will help explore the effect of any amount of breastfeeding versus no breastfeeding at all. In Methods/Definitions section, lines 141-142 we added the following sentence for further clarification: “These groups are not mutually exclusive”.

3. Line 149. How to distinguish EFF from Any FF if “Any FF” is without human milk?

Answer: Similar to our reply above in Point 2, “Any FF” includes all groups except “EBF” which allows us to explore the effect of less amounts of BF versus 100% exclusive BF. 

4. Line 218. There are six groups but why only conduct the analysis to compare three of them?

Answer: We ran comparisons for 3 different partitions of our sample across nutrition groups. In each case, the groups are mutually exclusive.

Comparison 1: “EBF/Predominant BF” vs “Mixed” vs “EFF”: This comparison is testing whether there is a dose-effect relationship between breast milk intake and the outcomes of interest. 

Comparison 2: “Any BF” vs “EFF”. As stated in our reply to comment 2 above, this comparison is exploring the effect of any amount of breastfeeding versus no breastfeeding (0%) on the outcomes of interest.

Comparison 3: “Any FF” vs “EBF”. As stated in our reply to comment 3 above, this comparison is exploring the effect of any amount of breastfeeding versus 100% exclusive breastfeeding on the outcomes of interest. 

Response to Reviewer 2:

Thank you for the valuable time you put to review our manuscript. Please find below our reply to each comment.

1. The abstract is too long. You can shorten the method section specifically the desription of the regression.

Answer: The abstract has been shortened as suggested. Please refer to the revised abstract.

2. Introducction: Please elaborate on the significance of the study.

Answer: We added more information on the clinical and policy significance of the study in the below paragraph. The clinical effectiveness of the intervention under study in improving breastfeeding rates was already established in the report of our trial [15]. However, scaling up of the intervention would require a cost-benefit analysis. In the Introduction section of the revised manuscript, we elaborated more on the current socio-economic situation in Lebanon to further show the significance of the study today: identifying cost-effective interventions that promote breastfeeding would be especially valuable today since Lebanon has been losing ground on most other factors that protect neonatal and child health. 

“While the effectiveness of the intervention on promoting EBF was already established in our trial [15], providing evidence that the intervention is also cost effective is essential for any scaling up of such an intervention. As the country loses ground on many other factors protecting child health (immunization, access to healthcare and medication, access to clean water) because of the economic crisis, identifying cost-effective interventions that impact protective behavioral factors such as breastfeeding is even more crucial”.

3. Introduction: Please add the hypothesis for your study. 

Answer: We added this paragraph at the end of the Introduction section: 

“We hypothesized that breastfeeding’s economic and infant-mother health benefits persist through the first two years. We therefore conducted this cost-benefit study to bridge the existing knowledge gap by assessing the direct and indirect costs of our multicomponent BF intervention at one, three, six, twelve, and 24 months postpartum, by including infant and maternal illnesses during these periods”. 

4. Methods, Participants: Please report the sample size calculations method. 

Answer: We added the following: 

“Participants were mothers who participated in the breastfeeding trial and completed at least one month of follow up (n=339). The trial had 90% power to detect a 10% difference in the 6-month EBF rate between the intervention and control groups, with 5% type 1 error and a potential 30% attrition rate. The sample size was estimated at 443 participants that were recruited from the antenatal clinics of two healthcare centers in Beirut, Lebanon”. 

5. Methods, Definitions: Please transfer it to the Introduction section. 

Answer: We respectfully disgaree with the kind reviewer’s suggestion as these definitions were articulated when we were putting together the data analysis plan of the study protocol in order to clearly categorize all possible infant nutritional outcomes. Hence, they are part of the Analysis plan which usually belongs to the Methods section. 

6. Methods, Data collection: Please report the validity and reliability of the data collection method and the outcomes. 

Answer: Thank you. The validity and reliability of the data collection methods including the outcomes were detailed in our published trial protocol and the published trial report (references 15, 21). We added the following sentences to the Data Collection section in the revised manuscript:

“Trained research assistants collected baseline and follow-up data, including nutrition, health, and cost data using standardized data collection forms. Nutrition data were adjudicated by lactation consultants. The primary investigator conducted weekly meetings with the research team to review the accuracy of collected data. Mothers were contacted for missing data or whenever data needed further scrutinization”. 

7. Line 154: Please clatify the education levels. 

Answer: Education levels were categorized as “less than university vs. university degree” in the revised manuscript.

8. Statistical analysis: Did you assess the assumption of the staistical test? Are there any violations to these assumptions?

Answer: The sample is large enough to safely assume that the central limit theorem holds, so that the tests of significance of ordinary least squares coefficient estimators from the regressions run for Table S1 are valid. Similarly, the sample size means running t-tests of differences in means in Tables S2 and S3 is warranted. Similarly, the sample size minimizes any concerns about the distributional assumption of normality needed to run ANOVA (Table S4). The assumptions needed to run a chi-squared test are all met (Table S4).

9. Baseline characteristics, Line 231: what did you do to control for the effect of these differences statistically?

Answer: The only significant differences in the baseline characteristics of the intervention and control groups were “the number of children” and “monthly household income”. These 2 variables were included in all multivariate regression models as predictors to adjust for them. In the Statistical Analysis section of the revised manuscript, we added the following sentence: 

“These variables were the the number of children and the monthly household income, both of which were entered in the regression models as predicros to adjust for their confounding effects.”

10. Comparisons by the type of infant nutrition: Please give more details about the type of statistical analysis name and variables included in each tests, nutrition.

Answer: We used the Independent Student t test to compare the means of continuous variables across the 2 nutrition groups shown in Tables S1, S2, and S3. Catgoriacl variables shown in Tables S1, S2, S3, and S4 were compared using Chi Square test. In Table S4, we used ANOVA to compare the continuous variables of the 3 nutrition groups. In the footnotes underneath each of these tables, we added the names of the statistical tests used for the different comparisons.

11. Discussion, Line 343: Please cite the few studies in the literature.

Answer: Done. These are refences [5, 25-28]. These are highlighted in the revised mansucript.

12. Discussion, Line 340: explain your findings. Why did you have such results?

Answer: We added the following sentence to explain the results of the first six months: “The higher intervention cost in the first six months can be due to the high number of home visits, costs of professional lactation support and BF education delivered to mothers in the intervention group”.

13. Discussion, Line 375: cite the Blanded et al. directly after the author not at the end of the sentence.

Answer: Done.

14. Discussion, Line 379: “this intervention” which intervention?

Answer: We removed “this intervention” and replaced it by “home-based BF counseling”.

15. Discussion, Lines 422 to 430: transfer to the introduction and keep a summary for it here.

Answer: Thank you for this suggestion. However, when we transferred this paragraph to the Introduction, it did not fit very well. The reason is that the studies mentioned in the transferred paragraph talk about the cost benefit of BF practice as a public health intervention, whereas the studies that are part of the Introduction with the knowledge gap are focusing on specific BF interventions delivered to a subgroup of the community. Hence, we decided to keep this paragraph in the Discussion where we compare our findings of infant’s BF health benefits to what has been published about its public health benefits. 

16. Limitations: You need to consider the effect of sample attrition.

Answer: Thank you. We added the following sentence to the Lmitations section: “It should also be noted that our sample suffered from attrition, with a total of 50 participants out of 339 withdrawing before month 24. Attrition is not uncommon in studies with long follow up, however, it is somewhat reassuring that the rate of attrition in this study is balanced across the two groups”.

We hope we have addressed all the comments of the kind reviewers in a clear manner, and revised the manuscript accordingly.

Thank you again for the thorough review of our paper.

Best wishes.

Sincerely, 

Mona Nabulsi, MD, MSc

Professor of Clinical Pediatrics

Department of Pediatrics and Adolescent Medicine

Faculty of Medicine

American University of Beirut

Beirut-Lebanon

P.O.Box: 113-6044/C8

E-mail: mn04@aub.edu.lb

---

## [Decision Letter · Decision Letter 1]

1 Jul 2024

Cost-benefit analysis of a multicomponent breastfeeding promotion and support intervention in a developing country

PONE-D-23-37306R1

Dear Dr. Mona%,

We’re pleased to inform you that your manuscript has been judged scientifically suitable for publication and will be formally accepted for publication once it meets all outstanding technical requirements.

Kind regards,

Kahsu Gebrekidan, Ph.D.

Academic Editor

PLOS ONE

Additional Editor Comments (optional):

Reviewers' comments:

Reviewer's Responses to Questions

**Comments to the Author**

1. If the authors have adequately addressed your comments raised in a previous round of review and you feel that this manuscript is now acceptable for publication, you may indicate that here to bypass the “Comments to the Author” section, enter your conflict of interest statement in the “Confidential to Editor” section, and submit your "Accept" recommendation.

Reviewer #1: All comments have been addressed

2. Is the manuscript technically sound, and do the data support the conclusions?

Reviewer #1: (No Response)

3. Has the statistical analysis been performed appropriately and rigorously? 

Reviewer #1: (No Response)

4. Have the authors made all data underlying the findings in their manuscript fully available?

Reviewer #1: (No Response)

5. Is the manuscript presented in an intelligible fashion and written in standard English?

Reviewer #1: (No Response)

6. Review Comments to the Author

Reviewer #1: (No Response)

7. PLOS authors have the option to publish the peer review history of their article (what does this mean?). If published, this will include your full peer review and any attached files.

Reviewer #1: No

---

## [Editor Report · Acceptance letter]

10 Jul 2024

PONE-D-23-37306R1 

PLOS ONE

Dear Dr. Nabulsi, 

I'm pleased to inform you that your manuscript has been deemed suitable for publication in PLOS ONE. Congratulations! Your manuscript is now being handed over to our production team.

Kind regards, 

on behalf of

Dr. Kahsu Gebrekidan 

Academic Editor

PLOS ONE